# Easy to learn hard to master - how to solve an arbitrary equation with PINN

**Alexander Hvatov**
ITMO University
St. Petersburg, Russia, 197101
`alex_hvatov@itmo.ru`

**Damir Aminev**
ITMO University
St. Petersburg, Russia, 197101
`aminevdom@itmo.ru`

**Nikita Demyanchuk**
ITMO University
St. Petersburg, Russia, 197101
`nikiniki@itmo.ru`

## Abstract

Physics-informed neural networks (PINNs) offer predictive capabilities for processes defined by known equations and limited data. While custom architectures and loss computations are often designed for each equation, the untapped potential of classical architectures remains to be seen. To make a comprehensive study, it is necessary to compare the performance of a given neural network architecture and loss formulation for different types of equations. This paper introduces an open-source framework for the unified handling of ordinary differential equations (ODEs), partial differential equations (PDEs), and their systems. We explore PINN applicability and convergence comprehensively, demonstrating its performance across ODEs, PDEs, ODE systems, and PDE systems.

## 1 Introduction

Based on a seminal paper describing the PINN architecture [1], researchers worldwide are trying to use it for different types of problem. It may also be considered an art to make PINN work for every given problem [2]. Therefore, many modifications appear for the classical problem statement with a dense network and for different architectures. Under scope are a loss function computation [3], data preprocessing and data handling overall [4], architecture changes [5] and other topics such as the use of more advanced combinations of training procedure and architecture[6].

It is always assumed that the function approximated with the neural network should also be close to a solution of a differential equation that partially or completely governs the process. However, apart from the classical approach, it is desirable to train differential equation coefficients also to be able to better reproduce the process. Therefore, different approaches exist for equation coefficients fine-tuning. Namely, we may tune equation coefficients in the form of constants or as variable coefficients using nested architectures such as DeepONet [7].

From the fundamental artificial intelligence point of view, we are interested in a common approximation ability in a given space. In the PINN-like application, we talk about Sobolev spaces and Sobolev-like training [8] procedures. Unlike the pure Sobolev training procedure, we basically use not only derivatives, but their combination as a whole equation, i.e. use a projective module instead of free one as algebraic foundation. In practice, we cannot move from an art of PINN creation to a stable working technique, since we could not assess which boundary-value problem solution can be approximated using PINNs with existing theory and tools. Therefore, it is convenient to have a tool

NeurIPS 2023 AI for Science Workshop.

that is able to solve different equations using the same toolset. Basic features of differential equation solving cover the DeepXDE framework [9].

However, it is not enough to explore all appearing features and their influence on the approximation ability. In the paper, we expand and describe the features that allow one to investigate the approximation ability of neural networks started for dense networks with more classical (without differentials) losses [10]. As PINN architectures, we chose two architectures: a dense network with activation and a single-layer linear network without activation. Further parameters are supported, including weak-form loss computation for an arbitrary set of basis functions. Furthermore, we support more (for ODE, PDE and their systems) adaptive loss weights than for only single PDEs in [11] based on Sobol indices instead of the neural tangent kernel and Fourier feature layers [12]. The main advantage is that all features are considered as equation-independent work as much as possible for ODE (ordinary differential equation), PDE (partial differential), and their systems.

The target application is to solve the equations that appear in the equation discovery process [13]. However, such a tool has wider applications and may be used as a separate research tool. We could already reach more than the $O(10^{-2})$ to $O(10^{-3})$ order of relative error that the usual PINNs reach [14] and with such a tool, we can study the sources of such error and make the neural network solver more applicable for practical applications. The global goal is to answer the question: "Which differential equations could be solved with a dense network, and what minimal error may be achieved?" Unlike many other approaches described, we make our research available as an open-source framework. [1]

## 2  Problem statement

As the Introduction follows, we aim to consider the classical PINN architecture without data term as a separate conventional equation solver. Namely, the classical PINN architecture uses the loss $\mathcal{L}$ in the form of Eq. 1.

$$\mathcal{L} = \lambda_1 \mathcal{L}_{data} + \lambda_2 \mathcal{L}_{op} + \lambda_3 \mathcal{L}_{bc} + \lambda_4 \mathcal{L}_{ic} \tag{1}$$

In Eq. 1 weights $\lambda_i \in R$ chosen expertly or using different principles that allow assessing the variance of each term like [11] or Sobol indices used in a described approach. The term $\mathcal{L}_{data}$ is usually a classical $l1$- or $l2$-loss used in dense network training. The terms $\mathcal{L}_{op}, \mathcal{L}_{bc}, \mathcal{L}_{ic}$ represent the equation ($op$) boundary ($bc$) and initial ($ic$) conditions. Different papers use different forms of losses, and the one used in the current paper is described below.

In this paper, we consider a problem without available solution data, that is, $\lambda_1 \equiv 0$. In this case, if the equation coefficients are considered fixed, the problem of loss minimization is equivalent to the approximation of the initial-boundary value problem solution with a given machine learning model.

### 2.1  Theoretical formulation

Generally, within an initial-boundary value problem part of the loss, we consider a differential equation system (ODE or PDE) that involves $p$ independent variables $x = (x^1, \dots, x^p)$ and $q$ dependent variables $\overrightarrow{u} = (u^1, \dots, u^q)$. The classical state of the boundary DEs system problem defined on the subdomain $x \in \Omega \subset R^p$ is:

$$\begin{aligned} S(\overrightarrow{u}) &= \overrightarrow{f} \\ b(\overrightarrow{u}) &= \overrightarrow{g} \end{aligned} \tag{2}$$

,where $S$ is a system operator, $b$ is an initial boundary operator (the initial and boundary conditions are not separated for brevity) on $\partial\Omega$, $\overrightarrow{f}$ is a source term and $\overrightarrow{g}$ is a boundary conditions value. Also, it is assumed that $S, b, \overrightarrow{f}, \overrightarrow{g}$ are defined in form when the boundary DEs problem is correct. Generally speaking, we want to find a converging series of solution candidates where the $n-th$ series term $\overrightarrow{u_n}$ is the numerical solution to the DE system boundary-value problem that satisfies Eq. 3.

---

[1] https://github.com/ITMO-NSS-team/torch_DE_solver

$$\lambda_2||S(\overrightarrow{u_n}) - \overrightarrow{f}|| + \lambda_3||b(\overrightarrow{u_n}) - \overrightarrow{g}|| \rightarrow_{n\to\infty} 0 \tag{3}$$

where $||.||$ is an arbitrarily chosen norm (usually, there is no need to use exotic norms, the $l2$ space is usually used for all solution forms). In the following, we show how the machine learning problem is formulated for the discrete grid.

## 2.2 Problem discretization. Formulation of machine learning problem.

We assume that the arbitrary machine learning model $\overrightarrow{u_*}(x; \theta)$ has a set of the parameters $\theta$. In the following, we mainly consider the dense neural network as a parametrized map $\overrightarrow{u_*}(x; \theta) : R^p \to R^q$. It should be mentioned that the proposed (PINN-like) numerical method is considered as mesh-free, usually uniform mesh is considered. The numerical minimization problem can be defined as Eq. 4 in $classical$ form.

$$\underset{\theta}{\operatorname{argmin}} \int_{\Omega} \|S(\overrightarrow{u_*}(x; \theta)) - f\|_i d\Omega + \lambda\|b(\overrightarrow{u_*}(x; \theta)) - g\|_j \tag{4}$$

Norms $\| \cdot \|_i$ and $\| \cdot \|_j$ can be arbitrarily chosen and $i$ and $j$ are placeholders (usually $i = l_2$ and $j = l_1$), but we mention that the proper connection with the induced space from the $D$ Sobolev space may give a better result for the solution of the DE system. We note that $\lambda$ is an arbitrarily chosen function (including a constant one), which will influence the convergence speed. In this case, there is no doubt that the solution of the optimization problem converges point-wise to the solution of the initial-boundary value problem Eq. 2.

If the form is $weak$ numerical minimization problem will be able to be defined as Eq. 5.

$$\underset{\theta}{\operatorname{argmin}} \sum_{\alpha=1}^{q} \int_{\Omega} [S(\overrightarrow{u_*}(x; \theta)^\alpha) - f^\alpha] \cdot \phi^\alpha \, d\Omega + \lambda\|b(\overrightarrow{u_*}(x; \theta)) - g\|_j \tag{5}$$

where $\overrightarrow{\phi} = (\phi^1, \dots, \phi^q)$ and $\phi^\alpha \in D(C^\infty(\Omega)$ with compact support) and the functions $\phi^\alpha, \alpha = (1, \dots, q)$ may be equal, including the case $\phi^1 = \dots = \phi^q = \phi$.

We note that from first glance there is no difference from a classical PINN problem statement. However, the details are important to be able to solve ODE and PDE equations and systems within one tool. The main features are as follows:

- Model form as a map;
- Division to 'Dirichlet'-like (prescribed values at selected points within the domain $\Omega$) and 'operator' boundary conditions (prescribed values of the arbitrary operator applied to a function) instead of classical division that is separate for every type of equations;
- Weak form incorporation.

Moving from formulae to code requires solving a series of engineering tasks. In the next section, we sketch the architecture of the solution.

## 2.3 Framework architecture.

The architecture of the solver is presented in Fig. 1 and in Appendix A in form of the pseudo-code, which includes three different methods (i.e. $mode$) for solving differential equations: `mat` based on a matrix (linear model without activation layers) and the others on neural network optimizations. One of the neural network optimization methods is `NN` which uses a finite difference scheme for derivative calculation, the other is `autograd` based on `pytorch` automatic differentiation algorithm, which is user-selectable.

- The module $initial\_data$ represents a user-defined module with crucial parameters such as $equation$, $boundary\_conditions$, $mode$, $grid$, and $model$. In addition, the following optional parameters are possible: $cache$, $lambda\_update$, $weak\_form$;

- The *preprocessing* module prepares *equation*, *boundary_conditions* to a unified form depending on the selected *mode*;
- The *solver* module combines sub-modules such as *cache*, *eval*, *derivative*, and *losses*. *Cache* allow you to use pre-trained models. *Derivative* provide differentiation depending on the selected *mode*, and *eval* applies the operator and boundary conditions to the model. Furthermore, *losses* compute the loss considering the selected type of loss function and adaptive $\lambda$ weights.

Eventually, the architecture of the solver is implemented so that it can be easily extended with new methods due to a unified structure (in terms of input/output of functions). For example, to define a new differential equation solution method, it is necessary to define a new solution method in the *preprocessing* module and the mechanism for determining the derivative in the *derivative* module. Another example, that module *models* realizes some simple neural network architectures (fully-connected and network with Fourier features), but it is possible to use other user-defined architectures. Moreover, in solver we do not stick to the neural networks; the proposed approach may be extended to an arbitrary parameterized model.

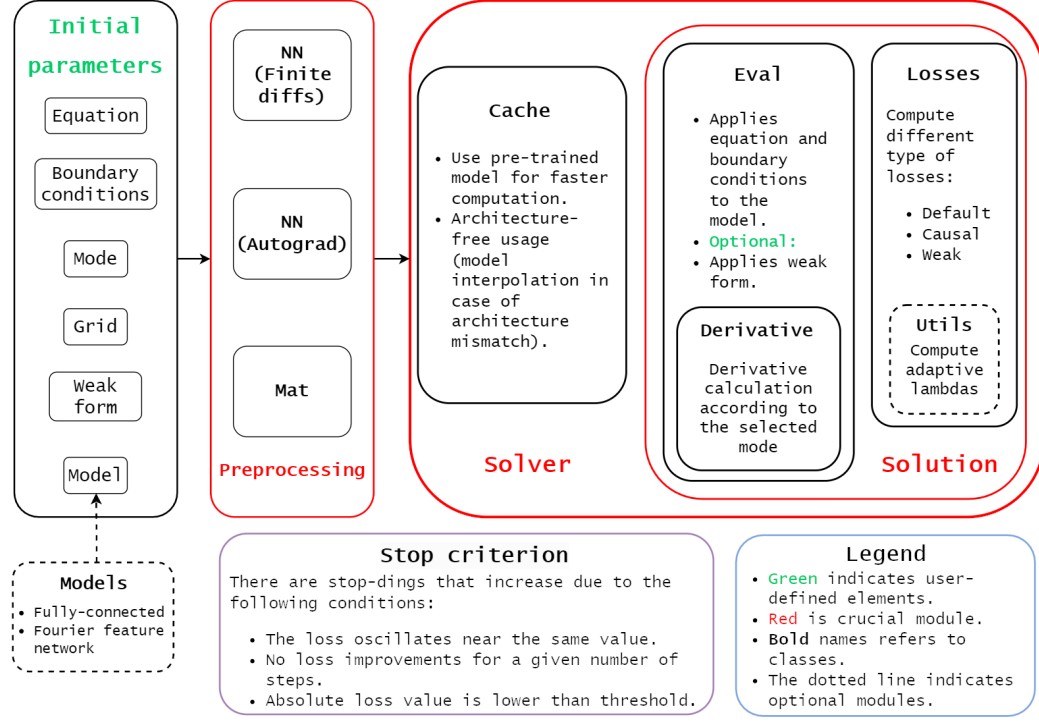

Figure 1: Solver architecture.

## 3 Experimental study

The following experiments show that the proposed approach could be applied to a wide range of initial and initial boundary value problems for single equation and their systems. As an example of approach applicability, the ODE, the system of ODE, the PDE, and the system of PDE were solved. The solution data was not used ($\lambda_1 \equiv 0$) to solve all the equations presented below. Graphs comparing the obtained solutions of equations with the reference solutions are given in the Appendix B. The experiments show that:

- adding points to the `grid` leads to a better solution, which evidences a convergency of the method;
- the obtained solutions may be more accurate than existing universal PINN solutions;

- using introduced re-training `cache` feature allows converging faster;

The formulations of the boundary problems used for the experiments and graphs of the $L_{op}$ (from Eq. 1) convergence for each `grid` are found in the Appendix C.

## 3.1 Van der Pol oscillator

The Van Der Pol oscillator is a non-conservative ODE oscilatting model with non-linear damping. It describes the relaxation-oscillation cycle in both the physical and biological sciences.

The numerical solution of the equation was obtained using the neural network architecture described in [15]. Generally, it is the same as a standard multilayer perceptron network (MLP), with the addition of two encoders and a minor modification in the forward pass. Specifically, the inputs are embedded into a feature space via two encoders and merged in each hidden layer of a standard MLP using a point-wise multiplication. As input, a Fourier feature embedded of the form $v(t) = (t, \sin(\omega t), \cos(\omega t), ..., \sin(m\omega t), cos(m\omega t))$ was fed into the neural network, where $\omega = \frac{2\pi}{L}$ and $m = 2, L = 10$.

The network included three hidden linear layers of 512 neurons, each with a hyperbolic tangent as an activation function. Fourier embeddings can be built for one-dimensional and multidimensional equations. Architecturally, this block is unified and can be used as a layer in neural networks of different architectures. The equation was solved using the `autograd` method and `Adam` optimizer. Experiments were carried out with different resolutions of the `grid` to quantify the precision of the solution obtained and the time spent. The results of the experiments are presented in Fig. 2, where $grid\_res$ indicates the number of discretization points for each independent variable (10 experiments were conducted for each $grid\_res$). The left part of Fig. 2 demonstrates that adding grid points improves accuracy (i.e. minimize root mean square error RMSE).

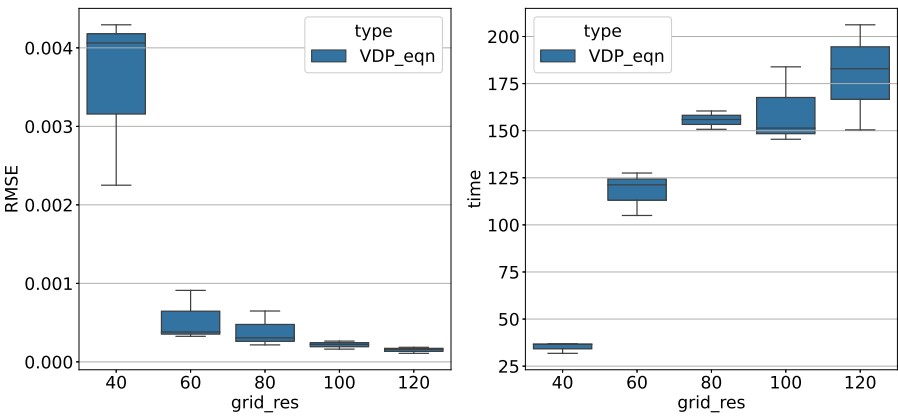

Figure 2: Results of the numerical solution to problem 6 for a different number of discretization points. RMSE (left) concerning the scipy.odeint solution and computation time in seconds (right). All experiments were performed with cache=False.

## 3.2 Lotka-Volterra equations

In science and engineering, problems are associated with more than one differential equation, so we have provided possibilities for solving ODE systems.The Lotka-Volterra system, also known as the predator-prey model, is a first-order nonlinear ordinary differential equation. These equations are often used to obtain the relationship of two interacting species.

The LV system was chosen because many authors resort to various tricks to solve it [16]. For example, the authors of DeepXDE use Fourier transformations of the neural network input to solve the system. They also transform the neural network output in a "hard constraint" manner to satisfy the initial conditions. All that is necessary is to specify a finite-difference template by which

numerical derivatives are determined and optimizer. For this case, the `central` numerical scheme $u'_c = \frac{1}{2}(u'_f + u'_b)$ was chosen, where $u'_f, u'_b$ are `forward` and `backward` schemes with $n = 3$ solution points (specified by the user as a hyperparameter) in each. To quantitatively assess the convergence of the algorithm, a series of experiments were carried out, which are shown in Fig. 3. The figure on the left shows that as the resolution of the grid increases (the increment decreases), the error in the solution decreases. It was also noted that using different initial guesses of the solution affected RMSE only in the third decimal place.

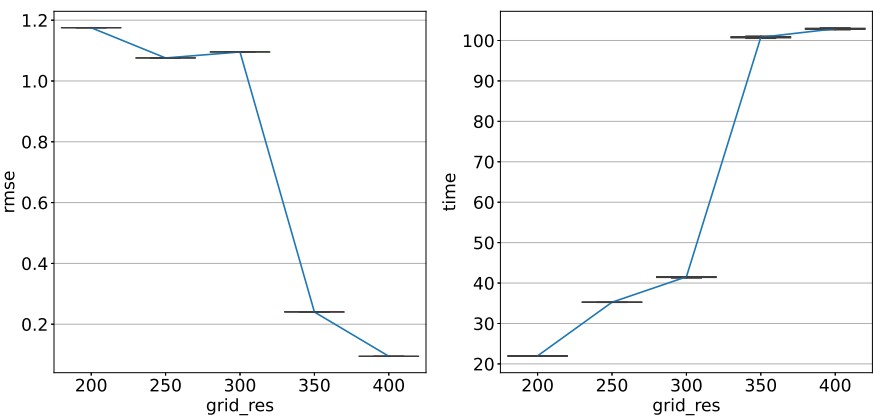

Figure 3: Results of the numerical solution to problem 7 for a different number of discretization points. RMSE (left) concerning the scipy.odeint solution and computation time is in seconds (right). All experiments were performed with cache=False.

## 3.3 Allen-Cahn equation

Having demonstrated the solver's ability to solve ordinary differential equations, it is necessary to analyze the numerical solutions obtained for the partial differential equation and compare them with the DeepXDE [9] approach. The Allen-Kahn equation was chosen, as an example as conventional PINN models are known to struggle [17].

The experiments were carried out to compare the accuracy and time of solving the Allen-Kahn equation with the proposed approach and DeepXDE. In both productions, the Adam optimizer was used. The DeepXDE implementation involves the use of a fully connected neural network with 3 hidden layers of 128 neurons and the transformation of neural network outputs ensuring compliance with the initial and boundary conditions. Specifically, the $x^2 cos(\pi x) + t(1 - x^2)u$ output transformation was used. In our implementation, three hidden layers of 128 neurons and a Fourier feature embedding ($m = 10, L = 2$) were used as input to the neural network.

Fig. 4 shows the results of the two approaches compared. In this experimental formulation, with an increase of the `grid` points number, the rate of error decrease (left figure) is more significant for the described approach. The time spent for the convergence of solutions in this experiment (right figure) shows that in the case of DeepXDE, the `grid` resolution has a greater impact on efficiency. The results obtained from the use of the two tools cannot be directly compared because different neural network architectures and different input/output transformations were realized.

However, under these conditions, the implementation proposed by our tool is more accurate and less time-consuming. Ensuring the fulfillment of boundary conditions in a "hard constraint" manner is not a general approach to solving equations, since one has to select a new transformation function in each case. Moreover, due to the variability of methods for solving equations implemented in the described approach, it is possible to compare and analyze solutions obtained by different methods. For example, the `mat` method solves this problem without additional data manipulation, achieving the $O(10^{-3})$ order of relative RMSE.

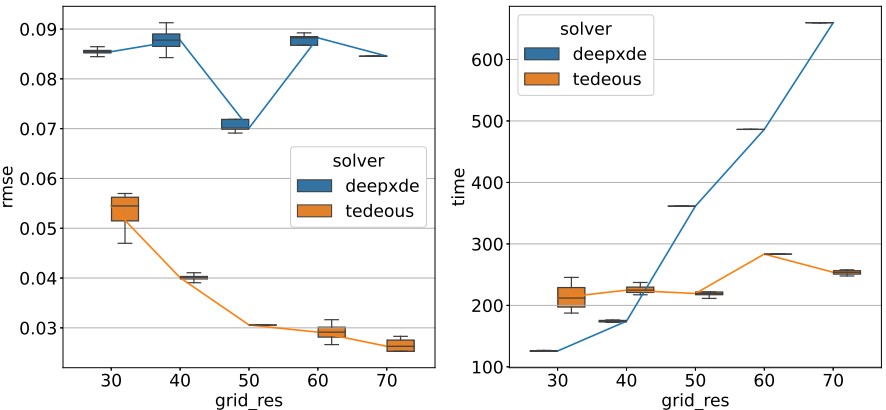

Figure 4: Results of the numerical solution to problem 8 for the proposed TEDEouS approach and DeepXDE. Solution time (right) and solution error (left). The reference solution is computed using Wolfram software. For each `grid_res`, the experiment is carried out ten times.

## 3.4 Nonlinear Schrodinger equation

The second example of solving PDE systems is the Schrodinger differential equation, which was solved as a system of two PDE for both real and imaginary parts just like at [1]. The nonlinear Schrodinger equation is a classical field equation that plays an important role in nonlinear wave theory, particularly in nonlinear optics.

The reference high-resolution data (`exact`) for comparison solutions were obtained using the spectral method with the $[256 \times 201]$ points grid.

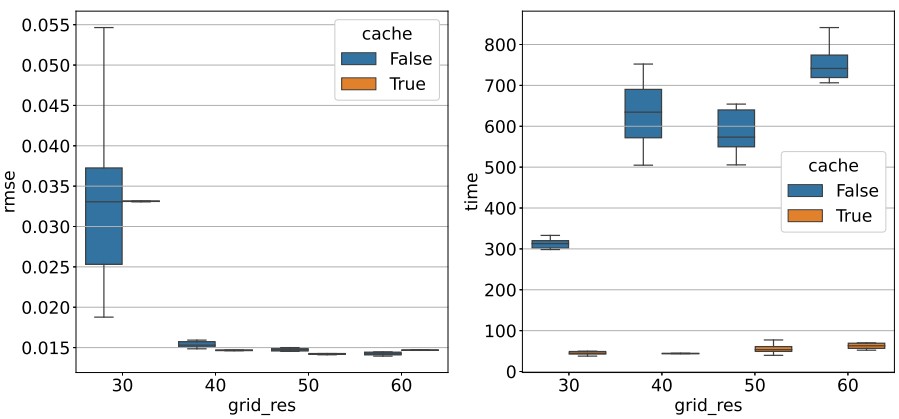

Figure 5: Schrodinger numerical solution error and computation time in seconds. Results with `cache=True` and `cache=False` are compared.

The time to solve all the previous examples was about 1-3 minutes, but when switching from single PDE equation to system, the time to solve the problem increases by a multiple. As described above, the proposed approach implements caching of trained neural networks. Using the cache allows to solve the optimization problem in less time, and a better initial approximation allows the optimization algorithm to converge more accurately with less variance.

Fig. 5 compares the error in solving the Schrodinger equation and the time spent for implementation with and without a `cache`. As you can see, the convergence time of the algorithm and the error variance are getting smaller than in realization without using a cache. The experiments were carried

out with a fully connected neural network with 5 hidden layers of 512 neurons, hyperbolic tangent activation function and `LBFGS` optimizer. The cache is implemented so that models trained by different methods are stored in a unified manner and can later be reused as initial guesses.

## 4  Conclusion

We propose a framework that allows us to solve ODE, PDE and their systems within a single framework. The main features are as follows:

- Differentiation algorithm choice - autograd and numerical one
- Single-layer model that allows to enhance the solution precision for most of the equations
- Weak form solution option
- Adaptive weights option

All options are available for different boundary-value problem formulations, even non-canonical ones. The purpose of the framework is to study the PINN convergence way and thus to make PINNs not an art but a technique. However, even in its current state, it may be used to solve different problems more precisely than existing neural network-based methods.

### Data and code availability

For review purposes, the experiments are available in the repository `https://github.com/ITMO-NSS-team/Solver-paper-for-AI4S`.

### Acknowledgments and Disclosure of Funding

This work was supported by the Analytical Center for the Government of the Russian Federation (IGK 000000D730321P5Q0002), agreement No. 70-2021-00141.

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

# A   Framework pseudocode

One of the neural network optimization methods is `NN` which uses a finite difference scheme for derivative calculation, the other is `autograd` based on `pytorch` automatic differentiation algorithm, which is user-selectable, as described in Algorithm 1 in lines 1-3. To reduce model training time, the solver provides a *cache* module (lines 5-6, algorithm 1) that allows you to use pre-trained models as an initial approximation. Lines 9-10 use modules *eval* (including the parameter *weak_form*) and *derivative* that are presented in Fig. 1. The *losses* module in line 11 contains several types of losses: default and weak (as described in Eqs. 4-5 and causal [18]. Lines 11-12 are updates to the $\lambda$ parameter in Eqs. 4-5.

---

**Algorithm 1:** Equation solver

---

**Input**     : grid, equation, boundary_conditions, model, mode
**Optional** : cache, optimizer, loss_type, lambda_update, weak_form
**Output**    : Trained model

1  **if** *mode == {NN, autograd, mat}* **then**
2  | $equation \leftarrow operator\_prepare(equation, mode)$
3  | $boundary\_conditions \leftarrow bnd\_prepare(boundary\_conditions, mode)$
4  **end if**
5  **if** *cache* **then**
6  | $model \leftarrow cache\_lookup()$
7  **end if**
8  **while** *stop criterion* **do**
9  | $equation \leftarrow operator\_compute(equation, model, weak\_form)$
10 | $boundary\_conditions \leftarrow bnd\_compute(boundary\_conditions, model)$
11 | $loss \leftarrow Losses.compute(equation, boundary\_conditions, lambda)$
12 | **if** *lambda update* **then**
13 | | $lambda \leftarrow Lambda.update(equation, boundary\_conditions, loss)$
14 | **end if**
15 **end while**
16 **return** $model$

---

# B Comparison of the obtained and reference solutions

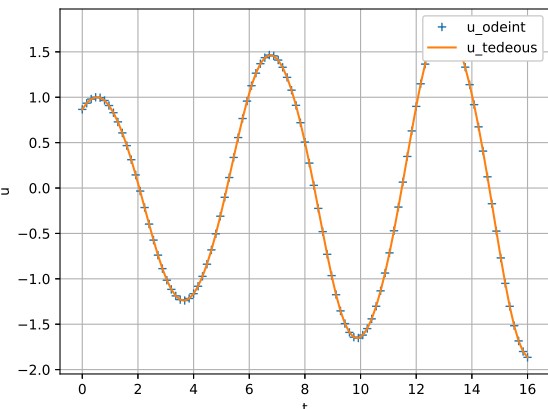

Figure 6: Van der Pol equation numerical solution (`grid_res=120`) compared with scipy.odeint solution.

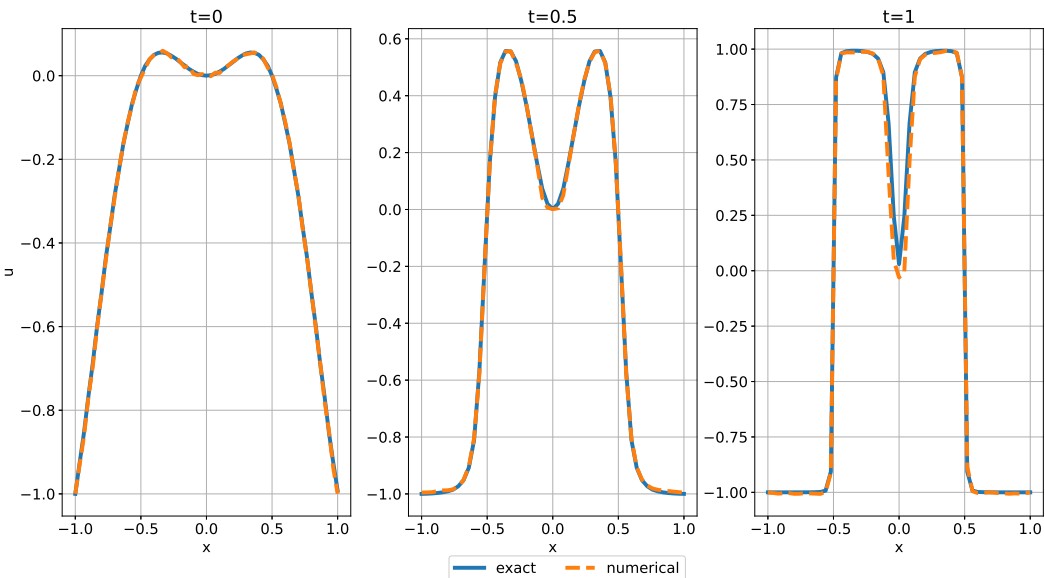

Figure 7: Allen-Cahn equation numerical solution with respect to the reference (wolfram) solution (with `grid_res=[60,60]`) at different moment of time.

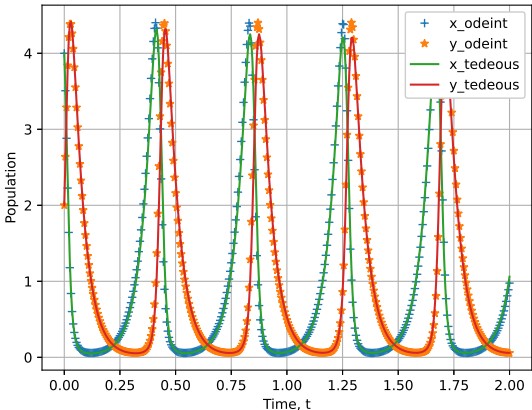

Figure 8: Lotka Volterra equations numerical solution (`grid_res=400`) compared with scipy.odeint solution.

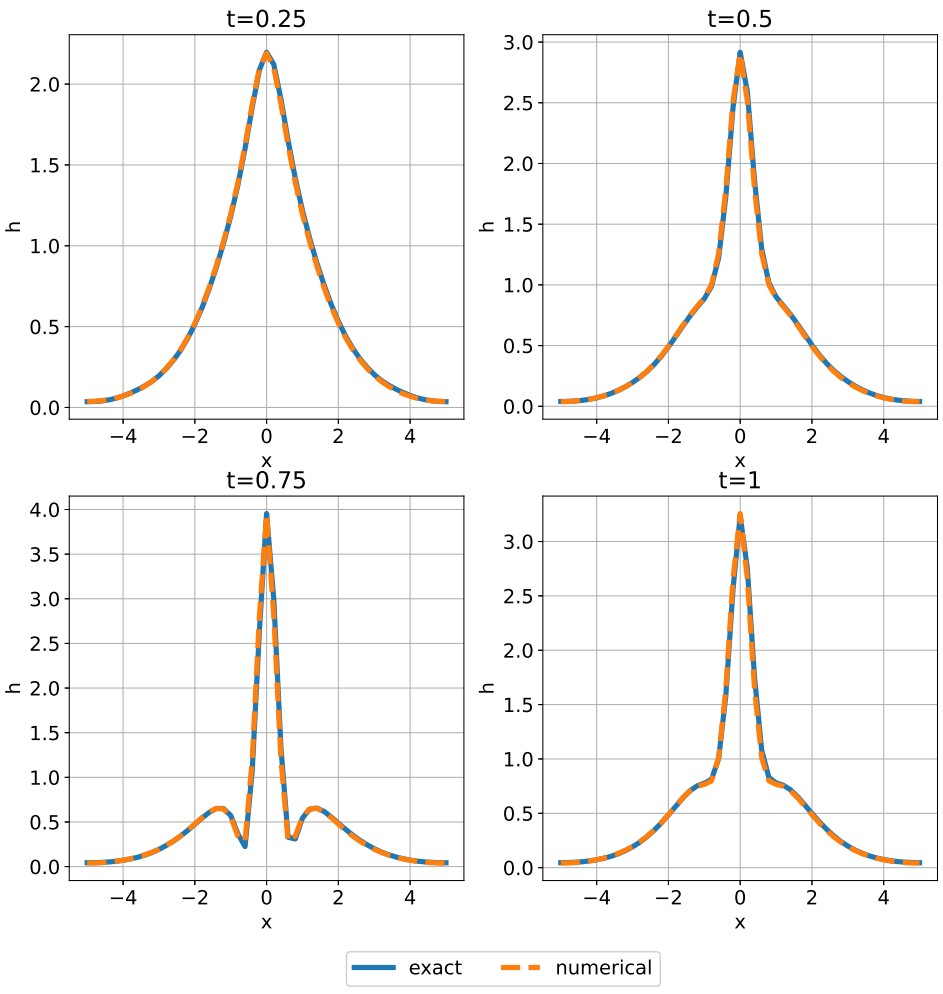

Figure 9: Schrodinger equation numerical solution (with `grid_res=[50,50]`) comparison with high-resolution data set.

## C  Boundary-value problem formulations

### C.1  Van der Pol oscillator

Eq. 6 is the second-order differential equation. When $\epsilon = 0$, this is a form of the simple harmonic oscillator. In this experiment, $\epsilon = 0.2$.

$$u'' + \epsilon(u^2 - 1)u' + u = 0$$
$$u'(t = 0) = 0.5$$
$$u(t = 0) = \sqrt{3}/2 \qquad (6)$$
$$t \in [0, 16]$$

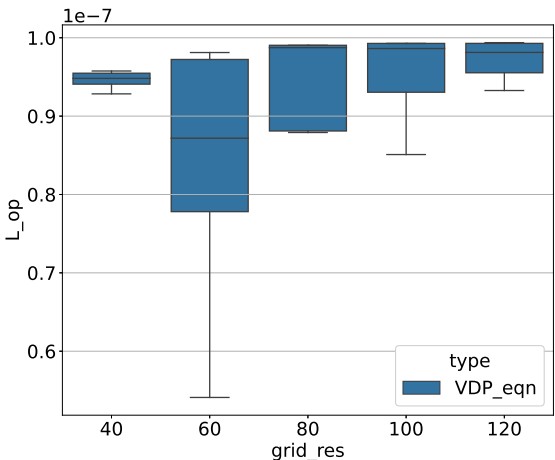

Figure 10: Pointwise convergence according to the $l2$ loss for the Van der Pol equation ($L_{op}$ term of Eq. 1). Results for a different number of discretization points.

### C.2  Lotka-Volterra equations

Each population changes through time according to Eq. 7:

$$\frac{\partial x}{\partial t} = x\alpha - yx\beta$$
$$\frac{\partial y}{\partial t} = -y\delta + yx\gamma$$
$$x(0) = x_0 = 4 \qquad (7)$$
$$y(0) = y_0 = 2$$
$$t \in [0, 2]$$

### C.3  Allen-Cahn equation

We consider the one-dimensional partial differential equation.

$$\frac{\partial u}{\partial t} = \alpha \frac{\partial^2 u}{\partial x^2} - 5u^3 + 5u$$
$$t \in [0, 1], x \in [-1, 1]$$
$$u(x, 0) = x^2 \cos(\pi x) \qquad (8)$$
$$u(t, -1) = u(t, 1)$$
$$\frac{\partial u}{\partial x}(t, -1) = \frac{\partial u}{\partial x}(t, 1)$$

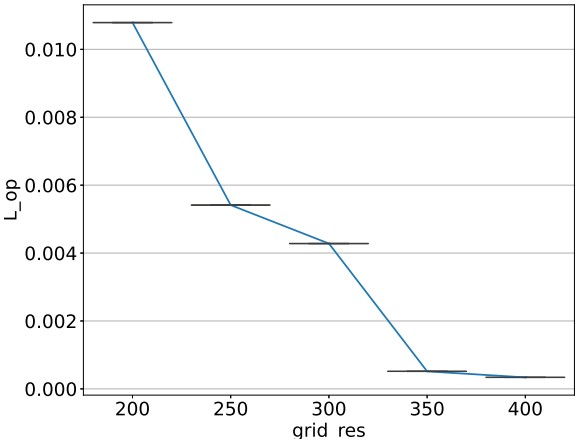

Figure 11: Pointwise convergence according to the $l2$ loss for the Lotka-Volterra equations ($L_{op}$ term of Eq. 1). Results for a different number of discretization points.

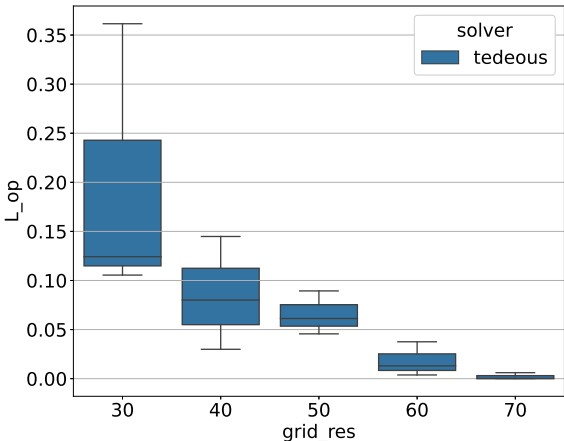

Figure 12: Pointwise convergence according to the $l2$ loss for the Allen-Cahn equation ($L_{op}$ term of Eq. 1). Results for a different number of discretization points.

### C.4 Schrodinger equation

The nonlinear Schrodinger equation with initial and boundary conditions are written as Eq. 9:

$$
\begin{aligned}
i\frac{\partial h}{\partial t} + \frac{1}{2}\frac{\partial^2 h}{\partial x^2} + |h^2|h &= 0 \\
h(x,0) &= 2\mathrm{sech}(x) \\
h(-5,t) &= h(5,t) \\
\frac{\partial h}{\partial x}(-5,t) &= \frac{\partial h}{\partial x}(5,t) \\
(x,t) \in [-5,5] \times [0,\pi/2] &= \Omega
\end{aligned}
\tag{9}
$$

Since $h$ is a complex-valued Eq. 9 could be represented as a PDE system. Let $h = u + iv$, then:

$$\frac{\partial u}{\partial t} + \frac{1}{2}\frac{\partial^2 v}{\partial x^2} + (u^2 + v^2)v$$
$$\frac{\partial v}{\partial t} + \frac{1}{2}\frac{\partial^2 u}{\partial x^2} + (u^2 + v^2)u$$
$$u(x,0) = 2sech(x), \quad v(x,0) = 0 \qquad (10)$$
$$u(-5,t) = u(5,t), \quad v(-5,t) = v(5,t)$$
$$\frac{\partial u}{\partial x}(-5,t) = \frac{\partial u}{\partial x}(5,t), \quad \frac{\partial v}{\partial x}(-5,t) = \frac{\partial v}{\partial x}(5,t)$$

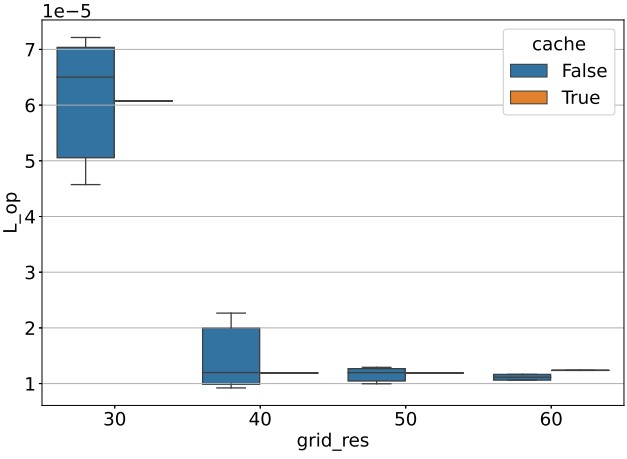

Figure 13: Pointwise convergence according to the $l2$ loss for the Schrodinger equation ($L_{op}$ term of Eq. 1). Results with `cache=True` and `cache=False` for different `grid_res` are compared.

