# OpenReview forum: "Easy to learn hard to master - how to solve an arbitrary equation with PINN"
_NeurIPS.cc/2023/Workshop/AI4Science — NeurIPS2023-AI4Science Poster_

### Official Review · Reviewer_dhuG · 2023-10-23
**Review of Easy to learn hard to master - how to solve an arbitrary equation with PINN**

**Rating:** 7
**Confidence:** 3

**Review:**

**1. Summary:**

The authors discuss the application and optimization of Physics-Informed Neural Networks (PINNs) in solving complex differential equations with known governing equations and limited data. They introduce a novel, comprehensive open-source module designed to facilitate a detailed comparative analysis between various neural network architectures and loss formulations.

**2. Strengths:**

**(a)** The authors' introduction of an open-source module for comparing PINN performances is a notable contribution.
**(b)** The paper not only discusses custom architectures but also explores classical architectures, offering a more comprehensive view.
**(c)** The incorporation of user-defined modules and the option to choose between autograd and numerical differentiation algorithms is nice feature.

**3. Weaknesses:**

Clarity in Experiments: The experimental section could be improved with a more detailed presentation of the setup, datasets, and specific results to enhance the readers' understanding.

**4. Detailed Comments:**

**Clarity and Quality of Writing:**
The paper is well-structured, but there could be improvements in the detailing of the experimental setup and results to ensure that readers can follow and reproduce the experiments effectively.

**Methodology:**
The authors employ a diverse range of methods, including a matrix-based linear model and neural network optimizations, to solve differential equations. The user-defined module, which is equipped with crucial parameters for various equation types and boundary conditions, is a highlight. The reviewer believes that a more in-depth explanation and visualization of the methods could be beneficial to improve the clarity.

**Experiments and Results:**
Experiments are designed for converging series of solution candidates for differential equations. While norms are used to evaluate solution forms, a more detailed presentation of the results, by including other physical metrics could be helpful for interpreting the results.

**Conclusion:**
The authors propose a unified framework. Despite some potential improvements, this work is a solid contribution to the field.

**5. Suggestions for Improvement:**

**(a)** Enhance the experimental section with visual representations to convey results more effectively.
For example, the authors could consider adding visualizations that help to explain the differential equations.
**(b)** Add additional comments interpreting the physical results. The authors used the norms as a metric to measure the results. However, some physical observables can be lacking for a better interpretation of the quality of the results.
**(c)** In section 3.4, the authors solve the Schrödinger equation, the review wonders why the authors focus on the nonlinear Schrödinger equation instead of the standard linear version. The linear version may sound simple, but the difficulty lies in the high dimensionality of many-body interactions. The reviewer suggests the authors change the title to nonlinear Schrödinger equation for the section if the authors have no interest in solving the standard linear version.
**(d)** The reviewer also suggests the authors add more references to existing works for each differential equation.

**6. Reproducibility:**

The authors have mentioned an open-source module, indicating a positive step towards reproducibility. However, the availability of datasets and specific implementation details may be further clarified to further improve the reproducibility of the results.

**7. Overall Evaluation:**

The paper is a valuable contribution to the field of differential equations solving using PINNs, with the introduction of a flexible and adaptable module being a significant strength. The reviewer suggests the paper be accepted.

**Additional comments:**
use a projective module instead of free one as algebraic foundation -> use a projective module instead of a free one as the algebraic foundation
oscilatting -> oscillating
fourier -> Fourier
Schrodinger -> Schrödinger

---

### Meta-Review · Area_Chair_jgkF · 2023-10-27

**Recommendation:** Accept (Poster)
**Confidence:** 3

**Metareview:**

The paper presents a valuable contribution to the field of differential equations solving using Physics-Informed Neural Networks (PINNs). It introduces an open-source module for comparing PINN performances, explores both custom and classical architectures, and incorporates user-defined modules, which are commendable strengths. However, there are concerns about the clarity of the experimental section and suggestions for improvements in methodology, result presentation, and reproducibility.

I recommend accepting the paper with the condition that the authors address the reviewers' suggestions to enhance the clarity, reproducibility, and interpretation of the results.